# Analysis of Combat in Sport JU-JITSU during the World Championships in Fighting Formula

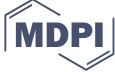

**Tadeusz Ambroży** [1], **Łukasz Rydzik** [1,*], **Wojciech Wąsacz** [1], **Zbigniew Małodobry** [2], **Wojciech J. Cynarski** [3], **Dorota Ambroży** [1] and **Andrzej Kędra** [1,*]

1   Institute of Sports Sciences, University of Physical Education, 31-571 Kraków, Poland; tadek@ambrozy.pl (T.A.); dorota.ambrozy@awf.krakow.pl (D.A.)
2   Podhale State Vocational University in Nowy Targ, 34-400 Nowy Targ, Poland; maloz@vp.pl
3   Institute of Physical Culture Studies, College of Medical Sciences, University of Rzeszow, 35-959 Rzeszów, Poland; ela_cyn@wp.pl
*   Correspondence: lukasz.rydzik@awf.krakow.pl (Ł.R.); aakedra@poczta.onet.pl (A.K.)

**Abstract:** Background: The observation and specialized analysis of confrontations in combat sports are fundamental for making corrections in training programs as well as for modifying individual technical–tactical profiles of athletes in such activities. These actions comprehensively assess the course of sports activities and ultimately inspire and guide the type of training in academies and sports clubs. The aim of the study was a general and detailed analysis of sport fighting in JU-JITSU during a top-tier tournament, in the fighting formula for the entire competition and for each weight category. Methods: The research material consisted of multimedia recordings of sports fights taken during the World JU-JITSU Championships in the fighting formula, Wroclaw 2016. A total of 229 tournament fights were analyzed in seven weight categories. For the purpose of evaluating the structure of the fight, a retrospective analysis of the recorded empirical material was conducted, and technical–tactical preparation (TTII) indicators were calculated, both in a global tournament context and for individual weight categories. Results: Of the 229 matches, more than half were decided by the advantage of technical points (58.52%) within the regulatory fight time, while in 74 clashes, victory was declared by Full Ippon (32.31%) before the designated fight time. Activity and attack effectiveness, as well as the number of technical points, were highest in the first part of the fight. Significant variations were observed in terms of activity, attack effectiveness, and point gains for all parts of the clash, and for effectiveness in the weight categories compared to the second part. The most frequently occurring penalty was the minor shido penalty, while the offense was the lack of fighting in the second part of the duel. The total fight time was 256 s, of which 144 s were effective fighting, and 112 s were breaks. Conclusions: The analysis of sports fight observations revealed that the majority of fights ended with a technical point advantage win (58.52%), with notable activity and attack effectiveness in part I. Middleweight fighters were most active early on, while heavyweight categories dominated later phases. Attack efficiency varied across weight categories. Penalties were predominantly minor (shido), and the total fight time included 144 s of effective fighting and 112 s of breaks.

**Keywords:** combat sports; fight analysis; technical–tactical indicators; sport Ju-Jitsu

## 1. Introduction

Ju-Jitsu is an ancient Japanese martial art derived from brutal hand-to-hand combat [1]. Historically, it is the "ancestor" of many martial arts disciplines, and it was the basis for the development of Judo and Aikido, among others [2]. The philosophy of Ju-Jitsu is multidimensional and assumes the comprehensive development of a person's physical and psychological aspects, in areas such as motor skills, psychology, technical–tactical, and utility [3].

In the sporting dimension, Ju-Jitsu as direct combat in the fighting formula is a cross-sectional and complete martial arts discipline, showcasing a rich range of technical–tactical training [4]. Specialized combat techniques are used in the fight, including various strikes with the upper limbs, kicks with the lower limbs, throws, takedowns, limb locks, chokes, and restraining positions. The goal of the competition is to dominate the opponent by ending the match before time (Full Ippon) or by technical point advantage [5]. The competition takes place on three levels (3 parts of the match). The fight starts from a standing position where competitors exchange strikes and kicks (Part I). When it comes to grappling, they struggle to bring each other to the ground through throws and takedowns (Part II). Ground combat using holds, chokes, and joint locks is also allowed (Part III) [5]. Competitors compete in traditional attire (judogi) and protectors, divided into weight categories according to IJJF (International Ju-Jitsu Federation) regulations [5].

Thematic literature suggests that the largest share in the preparation profile of martial arts athletes is technical–tactical (31.3%) and physical (structural–functional 28.4%), with theoretical preparation (25.5%) and psychological (14.8%) taking a smaller share [2]. Scientific observation and subsequent retrospective, detailed analysis of confrontations in martial arts contain multi-dimensional stimuli. These measures enable a technical–tactical diagnosis of training trends of a given nation, as well as recognizing the individual or group characteristics of competitors' competitive activities (differentiation by weight category, level of sports achievements, or discipline formula) [6,7]. Such measures comprehensively assess the course of sporting activities and ultimately inspire and determine the type of training in sports clubs [8]. This type of scientific activity and its impact on the broad understanding of the optimization of the training process can be observed in many martial arts disciplines [9–11].

Therefore, the issue of technical–tactical preparation seems to be highly relevant in the case of Ju-Jitsu (fighting formula), with a cross-sectional profile of the combat plan. Previous scientific reports precisely analyzed the time and material structure of the conduct of the fight [12] and its significance for elite athletes and coaches [13,14]. The effectiveness of techniques that occur in the first part of the fight was also studied [15,16]. Analyzing the Ju-Jitsu fights at the Junior World Championships in Bucharest in 2013, the most frequently used techniques in the various parts of the fight were demonstrated [13–15]. In the research by Ambroży and co-authors in 2021, technical–tactical indicators were used to assess the entire clash without dividing it into individual parts of the fight. Moreover, based on empirical observation, it was noticed that, in the first part, the confrontation of fighters in the fighting formula resembles standing, striking, and kicking martial arts (Karate, Kickboxing), while in the second and third parts, the fight is similar to Judo [17]. Based on the mentioned literature and player–coach empiricism, in assessing sport Ju-Jitsu confrontations, it is necessary to analyze in detail such elements as the material and time structure of sports fighting. Due to the cross-sectional nature of the clash (standing, grappling, ground), success is determined by comprehensive technical–tactical preparation, the effectiveness of the performed techniques, and mistakes made by the opponent [11,18]. The condition for planning an effective training program aimed at achieving sports mastery is a precise analysis of the fight structure with the recognition of the requirements it imposes on athletes [19,20]. Moreover, according to Bocioaca, the factor that decides victory in an evenly matched duel between rivals with similar motor and technical profiles is the fight tactics [21]. Therefore, achieving sports mastery is possible thanks to proper technical–tactical preparation [22].

Hence, the question arises, what specific aspects of technique and tactics should be developed and improved in sport Ju-Jitsu to optimally support the realization of training and competition goals? The complexity of the training system for technical–tactical differentiation in the various parts of the fight poses a great challenge for coaches and athletes in this discipline. In available domestic and foreign literature, a deficit of scientific studies was identified, taking into account the comprehensive problems of the technical–tactical sphere for top-level competitions. The aim of this study was to provide a general and detailed

analysis of Ju-Jitsu sports fighting during the World Championships, in the fighting formula for all competitions, with particular emphasis on technical–tactical efficiency, and taking into account individual weight categories. The results of this study should determine the desired direction of action for theorists and practitioners to improve the quality of coaching control and thus further development of this discipline. The novelty of this study is the use of PPT indicators with division into individual parts of the fight. This is important for the training process in Ju-Jitsu because the technical and motor structures of the individual parts differ.

## 2. Materials and Methods

### 2.1. Participants and Events

The research material consists of multimedia recordings of sports fights taken during the Ju-Jitsu Seniors World Championship (fighting formula), which took place in Wroclaw on 25 and 26 November 2016. A total of 132 athletes from 49 countries participated, representing the top of world Ju-Jitsu. The athletes fought 229 matches in different weight categories (Table 1). The competition was conducted according to IJJF (International Ju-Jitsu Federation) rules.

**Table 1.** Summary of weight categories, number of athletes, and matches.

| Weight Category | Number of Athletes | Number of Fights |
|:---:|:---:|:---:|
| 56 kg | 10 | 15 |
| 62 kg | 23 | 41 |
| 69 kg | 24 | 43 |
| 77 kg | 22 | 39 |
| 85 kg | 29 | 53 |
| 94 kg | 13 | 21 |
| +94 kg | 11 | 17 |
| Total | 132 | 229 |

### 2.2. Research Design

Details of the experimental program are presented in Figure 1.

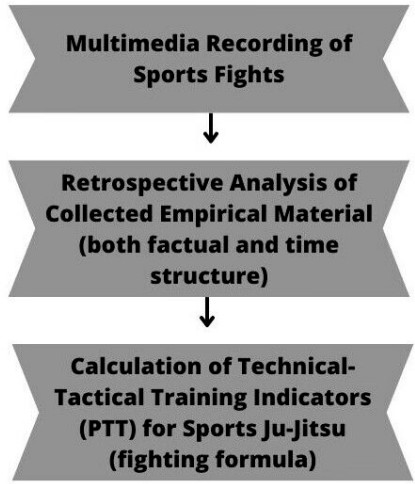

**Figure 1.** Experimental design.

### 2.3. Research Procedures

Four specialized Sony HDR-PJ6203 digital cameras (Sony, Tokyo, Japan) were used for video recording, covering the entire fighting area where the sports matches were held.

The camera setup allowed for continuous observation of the fighting athletes, referees, and the scoreboard. All matches from the elimination part to the medal rounds were recorded. Cameras were active from the start of the elimination matches to the end of the final matches. Full matches were recorded, including all breaks that occurred during their duration. Breaks in the competition due to the schedule were not recorded. Actions in the technical–tactical sphere were noted on a specialized observation sheet. A team of four Ju-Jitsu sports experts (three master class coaches and one internationally qualified referee) conducted a retrospective analysis of the recorded empirical material.

To profile the specific course of competition in Ju-Jitsu and its individual fighting parts and weight categories, both factual and time structures were analyzed.

In assessing the factual structure, offensive actions were analyzed both qualitatively and quantitatively. The type of match conclusion (before time vs. technical points advantage), type of attack (effective vs. ineffective), referee score (Wazaari—1 point; Ippon—2 or 3 points), and penalties (Shido, Chui, Hansoku-make) were noted at different parts of the fight (Part I: Atemi-waza—striking while standing, Keri-waza—kicks; Part II: Nage-waza—throws, takedowns, and groundwork; Part III: Ne-waza—groundwork involving holds, chokes, and joint locks). The percentage share and averages of selected parameters in the fight were then calculated.

For the time structure analysis, each match was divided into three active–passive segments. Using a stopwatch with an accuracy up to 1/100th of a second, the following times were measured:

1.  Total Time (entire duration of the fight, including breaks)—measured from the referee's Hajime command (start of the fight) to the Matte command (signaling the end of the fight).
2.  Break Time (breaks occurring between efforts in the fight)—measured from the Matte command (interrupting the competition) to Hajime (resumption of the match).
3.  Effective Fight Time (difference between total time and breaks)—measured from the Hajime command (start or resumption of the match) to Matte (pause or end of the competition).

### 2.4. Analysis of the Fight: Technical–Tactical Training Indicators (TTITTI)

The motivation for determining the model technical–tactical profile in sports Ju-Jitsu, based on observed activity in fights and using the specialized formulas presented below, was to calculate indicators characterizing the level of technical–tactical training. In scientific research, this strategy is often used in combat sports to assist coaching control and optimize the broad understanding of the training process [23].

1.  Attack Effectiveness in the First Part of the Fight
    **SaF1= ((n1 × 1) + (n2 × 2))/N**
    n1—number of attacks assessed as Waza-ari (1 point)
    n2—number of attacks assessed as Ippon (2 points)
    1, 2—point values for successful attacks
    N—total number of observed fights

2.  Attack Effectiveness in the Second Part of the Fight
    **SaF2 = ((n1 × 1) + (n2× 2))/N**
    n1—number of attacks assessed as Waza-ari (1 point)
    n2—number of attacks assessed as Ippon (2 points)
    1, 2—point values for successful attacks
    N—total number of observed fights

3.  Attack Effectiveness in the Third Part of the Fight
    **SaF3 = ((n1 × 1) + (n2 × 2) + (n3 × 3))/N**
    N1—number of attacks assessed as Waza-ari (1 point)
    n2—number of attacks assessed as Ippon (2 points)
    n3—number of attacks assessed as Ippon (3 points)

1, 2, 3—point values for successful attacks
N—total number of observed fights

4.  Attack Efficiency in the First Part of the Fight
    **EaF1 = (number of successful attacks/total number of attacks) × 100**

5.  Attack Efficiency in the Second Part of the Fight
    **EaF2 = (number of successful attacks/total number of attacks) × 100**

6.  Attack Efficiency in the Third Part of the Fight
    **EaF3 = (number of successful attacks/total number of attacks) × 100**

7.  Attack Activity in the First Part of the Fight
    **AaF1 = (∑a)/n**
    ∑a—sum of attacks
    n—number of fought matches

8.  Attack Activity in the Second Part of the Fight
    **AaF2 = (∑a)/n**
    ∑a—sum of attacks
    n—number of fought matches

9.  Attack Activity in the Third Part of the Fight
    **AaF3 = (∑a)/n**
    ∑a—sum of attacks
    n—number of fought matches

10. Referee Penalty Effectiveness Indicator
    **Sk = ((k1 × 1) + (k2 × 2))/N**
    k1—minor penalty "shido" (1 point)
    k2—moderate penalty "chui" (2 points)
    1, 2—point values for penalties
    N—total number of observed fights

*2.5. Statistical Analysis*

Statistical analysis of the collected data was conducted using Statistica software by StatSoft (version 13.1, StatSoft, Kraków, Poland). The following basic descriptive statistics were calculated: arithmetic mean, median, minimum and maximum values, the level of the first and third quartiles, standard deviation, and the coefficient of variation. To evaluate the significance of the differences between the variables, non-parametric Mann–Whitney U-Test and Kruskal–Wallis ANOVA were used, fulfilling the requirements of the nature of the variables being studied, which were verified using the Shapiro–Wilk test.

**3. Results**

The surveyed group of 132 competitors had a total of 229 sporting bouts. It was shown that in JU-JITSU, over half of the confrontations (58.52%) ended with a victory due to a technical points advantage within the full duration of the regulation bout time, i.e., 134 bouts. Of the observed fights, 95 (41.49%) ended before time, of which 74 (32.31%) were won by Full Ippon, while in 21 (9.17%), victory by one of the competitors was announced by the referees due to injury or refusal to continue the fight.

As can be seen in Table 2, the highest activity and effectiveness of attack were observed in Part I of the fight. Attack effectiveness took place in Part II of the fight and was associated with successfully transitioning the confrontation to the ground level. Detailed values of technical–tactical training indicators for all competitors participating in the World JU-JITSU Championships, Wrocław 2016, are presented in Table 2.

**Table 2.** Basic Descriptive Statistics of Activity, Effectiveness, and Efficiency of Attack in Different Parts of the Fight.

| Indicator | N | $\bar{x}$ | Me | Min | Max | $Q_1$ | $Q_3$ | SD | V |
|---|---|---|---|---|---|---|---|---|---|
| Activity Part I | 132 | 9.80 | 9.71 | 1 | 20 | 7.67 | 11.71 | 3.30 | 33.71 |
| Activity Part II | 132 | 1.57 | 1.50 | 0 | 4 | 1.00 | 2.00 | 0.92 | 58.51 |
| Activity Part III | 132 | 0.59 | 0.50 | 0 | 2 | 0.00 | 1.00 | 0.53 | 89.15 |
| Effectiveness Part I | 132 | 6.17 | 6.13 | 0 | 13 | 4.13 | 7.88 | 2.91 | 47.19 |
| Effectiveness Part II | 132 | 1.04 | 1.00 | 0 | 4 | 0.33 | 1.60 | 0.83 | 79.98 |
| Effectiveness Part III | 132 | 0.33 | 0.00 | 0 | 2 | 0.00 | 0.67 | 0.45 | 138.75 |
| Efficiency Part I | 132 | 46.21 | 46.33 | 0 | 77.41 | 39.34 | 52.86 | 13.49 | 29.19 |
| Efficiency Part II | 121 | 48.43 | 50.00 | 0 | 100.00 | 28.57 | 72.73 | 29.55 | 61.01 |
| Efficiency Part III | 98 | 46.50 | 50.00 | 0 | 100.00 | 0.00 | 75.00 | 35.92 | 77.26 |

N, number of subjects; x, arithmetic mean; Me, median; Min, minimum value; Max, maximum value; $Q_1$, lower quartile; $Q_3$, upper quartile; SD, standard deviation; V, coefficient of variation.

From the conducted comparative analyses, it appears that the average values of activity and effectiveness of the attack show significant variations in different parts of the fight. This trend was not observed in the case of attack efficiency, where the averages of individual parts are similar (Tables 2 and 3).

**Table 3.** Comparative Summary of Technical–Tactical Indicators in Three Parts of the Fight (Mann–Whitney U Test).

| Indicator | $p$ |
|---|---|
| Activity Part I vs. Activity Part II | 0.0000 * |
| Activity Part II vs. Activity Part III | 0.0000 * |
| Activity Part I vs. Activity Part III | 0.0000 * |
| Effectiveness Part I vs. Effectiveness Part II | 0.0002 * |
| Effectiveness Part II vs. Effectiveness Part III | 0.0000 * |
| Effectiveness Part I vs. Effectiveness Part III | 0.0000 * |
| Efficiency Part I vs. Efficiency Part II | 0.7411 |
| Efficiency Part II vs. Efficiency Part III | 0.6586 |
| Efficiency Part I vs. Efficiency Part III | 0.8806 |

* $p < 0.05$.

In the assessment of technical–tactical performance in individual weight categories, the highest average activity value for Part I was recorded in the up to 77 kg category, while for Parts II and III, it was recorded in the up to 85 kg category. The lowest values for Parts I and II were recorded in the heaviest weight category, above 94 kg, while for activity in Part III, it was recorded in the up to 62 kg category (Table 4).

**Table 4.** Basic Descriptive Statistics of Attack Activity in the Three Parts of the Fight, Divided by Weight Categories.

| Indicator | | 56 kg | 62 kg | 69 kg | 77 kg | 85 kg | 94 kg | +94 kg | Total |
|---|---|---|---|---|---|---|---|---|---|
| | $\bar{x}$ | 9.28 | 9.78 | 9.52 | 10.68 | 10.04 | 9.86 | 8.45 | 9.80 |
| | N | 10 | 24 | 24 | 22 | 29 | 12 | 11 | 132 |
| | SD | 2.92 | 3.70 | 3.21 | 3.26 | 3.00 | 3.91 | 3.40 | 3.30 |
| | Min | 3.50 | 2.50 | 6.00 | 5.50 | 5.00 | 4.00 | 1.00 | 1.00 |
| Activity Part I | Max | 14.00 | 15.00 | 18.67 | 20.00 | 16.14 | 16.50 | 13.33 | 20.00 |
| | $Q_1$ | 8.00 | 7.00 | 7.58 | 9.00 | 8.25 | 7.00 | 7.00 | 7.67 |
| | Me | 9.75 | 10.25 | 8.71 | 10.90 | 10.00 | 9.60 | 8.00 | 9.71 |
| | $Q_3$ | 10.67 | 12.70 | 10.83 | 11.67 | 11.25 | 11.75 | 11.25 | 11.71 |

**Table 4.** *Cont.*

| Indicator | | 56 kg | 62 kg | 69 kg | 77 kg | 85 kg | 94 kg | +94 kg | Total |
|---|---|---|---|---|---|---|---|---|---|
| Activity Part II | $\bar{x}$ | 1.42 | 1.27 | 1.49 | 1.81 | 1.87 | 1.69 | 1.16 | 1.57 |
| | N | 10 | 24 | 24 | 22 | 29 | 12 | 11 | 132 |
| | SD | 0.63 | 1.00 | 0.98 | 0.88 | 0.66 | 1.28 | 0.86 | 0.92 |
| | Min | 0.50 | 0.00 | 0.00 | 0.00 | 0.50 | 0.00 | 0.00 | 0.00 |
| | Max | 2.33 | 3.50 | 3.67 | 3.50 | 3.50 | 4.00 | 2.00 | 4.00 |
| | $Q_1$ | 1.00 | 0.45 | 1.00 | 1.33 | 1.50 | 0.50 | 0.00 | 1.00 |
| | Me | 1.42 | 1.00 | 1.27 | 1.78 | 2.00 | 1.83 | 1.50 | 1.50 |
| | $Q_3$ | 1.67 | 1.83 | 1.75 | 2.33 | 2.25 | 2.50 | 2.00 | 2.00 |
| Activity Part III | $\bar{x}$ | 0.59 | 0.50 | 0.51 | 0.59 | 0.70 | 0.63 | 0.68 | 0.59 |
| | N | 10 | 24 | 24 | 22 | 29 | 12 | 11 | 132 |
| | SD | 0.58 | 0.49 | 0.48 | 0.62 | 0.44 | 0.46 | 0.78 | 0.53 |
| | Min | 0.00 | 0.00 | 0.00 | 0.00 | 0.00 | 0.00 | 0.00 | 0.00 |
| | Max | 1.67 | 1.80 | 1.50 | 2.00 | 1.60 | 1.60 | 2.00 | 2.00 |
| | $Q_1$ | 0.00 | 0.00 | 0.00 | 0.00 | 0.33 | 0.34 | 0.00 | 0.00 |
| | Me | 0.50 | 0.50 | 0.37 | 0.33 | 0.67 | 0.58 | 0.50 | 0.50 |
| | $Q_3$ | 1.00 | 0.71 | 0.71 | 1.00 | 1.00 | 1.00 | 1.67 | 1.00 |

Regarding the effectiveness of the attack in Part I, the highest indicators were observed in the lowest weight category, up to 56 kg. In contrast, higher effectiveness in the fight is characterized by heavier competitors, i.e., up to 85 kg in Part II and above 94 kg in Part III. The lowest effectiveness was recorded in the above 94 kg category for Part I, up to 69 kg for Part II, and 77 kg for Part III (Table 5).

**Table 5.** Basic Descriptive Statistics of Attack Effectiveness in the Three Parts of the Fight, Divided by Weight Categories.

| Indicator | | 56 kg | 62 kg | 69 kg | 77 kg | 85 kg | 94 kg | +94 kg | Total |
|---|---|---|---|---|---|---|---|---|---|
| Effectiveness Part I | $\bar{x}$ | 7.28 | 6.45 | 5.09 | 6.65 | 6.59 | 6.14 | 4.90 | 6.17 |
| | N | 10 | 24 | 24 | 22 | 29 | 12 | 11 | 132 |
| | SD | 3.03 | 3.13 | 2.37 | 3.16 | 2.68 | 3.17 | 2.86 | 2.91 |
| | Min | 2.00 | 1.00 | 1.00 | 2.00 | 2.00 | 0.00 | 0.00 | 0.00 |
| | Max | 12.00 | 13.00 | 12.00 | 13.00 | 13.00 | 11.25 | 8.00 | 13.00 |
| | $Q_1$ | 5.00 | 4.75 | 3.83 | 4.33 | 5.00 | 3.86 | 3.50 | 4.13 |
| | Me | 7.50 | 6.00 | 4.42 | 6.58 | 7.00 | 6.67 | 5.33 | 6.13 |
| | $Q_3$ | 8.00 | 8.63 | 6.58 | 8.20 | 8.50 | 8.04 | 7.50 | 7.88 |
| Effectiveness Part II | $\bar{x}$ | 1.21 | 1.05 | 0.80 | 0.90 | 1.32 | 0.99 | 0.98 | 1.04 |
| | N | 10 | 24 | 24 | 22 | 29 | 12 | 11 | 132 |
| | SD | 0.98 | 0.80 | 0.68 | 0.77 | 0.92 | 0.99 | 0.72 | 0.83 |
| | Min | 0.00 | 0.00 | 0.00 | 0.00 | 0.00 | 0.00 | 0.00 | 0.00 |
| | Max | 3.33 | 2.67 | 2.00 | 2.80 | 4.00 | 3.25 | 2.00 | 4.00 |
| | $Q_1$ | 0.50 | 0.33 | 0.00 | 0.20 | 0.67 | 0.17 | 0.00 | 0.33 |
| | Me | 1.00 | 1.23 | 0.88 | 0.88 | 1.00 | 0.75 | 1.00 | 1.00 |
| | $Q_3$ | 1.50 | 1.71 | 1.27 | 1.50 | 2.00 | 1.55 | 1.67 | 1.60 |
| Effectiveness Part III | $\bar{x}$ | 0.42 | 0.35 | 0.23 | 0.21 | 0.37 | 0.22 | 0.62 | 0.33 |
| | N | 10 | 24 | 24 | 22 | 29 | 12 | 11 | 132 |
| | SD | 0.49 | 0.55 | 0.33 | 0.39 | 0.43 | 0.25 | 0.66 | 0.45 |
| | Min | 0.00 | 0.00 | 0.00 | 0.00 | 0.00 | 0.00 | 0.00 | 0.00 |
| | Max | 1.33 | 2.00 | 1.00 | 1.40 | 1.50 | 0.67 | 1.67 | 2.00 |
| | $Q_1$ | 0.00 | 0.00 | 0.00 | 0.00 | 0.00 | 0.00 | 0.00 | 0.00 |
| | Me | 0.25 | 0.00 | 0.00 | 0.00 | 0.33 | 0.14 | 0.50 | 0.00 |
| | $Q_3$ | 0.67 | 0.58 | 0.50 | 0.20 | 0.67 | 0.45 | 1.33 | 0.67 |

The highest values of attack efficiency in Parts II and III were shown by competitors in the above 94 kg category. The opposite trend was observed in Part I, where the highest

indicators were recorded in the lowest weight category, up to 56 kg. The least efficiency characterized the up to 69 kg category (Part I) and the up to 77 kg category (Parts II and III) (Table 6).

**Table 6.** Basic Descriptive Statistics of Attack Efficiency in the Three Parts of the Fight, Divided by Weight Categories.

| Indicator | | 56 kg | 62 kg | 69 kg | 77 kg | 85 kg | 94 kg | +94 kg | Total |
|---|---|---|---|---|---|---|---|---|---|
| Efficiency Part I | $\bar{x}$ | 58.79 | 50.46 | 39.67 | 43.92 | 49.16 | 42.92 | 40.15 | 46.21 |
| | N | 10 | 24 | 24 | 22 | 29 | 12 | 11 | 132 |
| | SD | 11.68 | 10.06 | 10.93 | 11.08 | 9.78 | 14.97 | 24.23 | 13.49 |
| | Min | 39.29 | 33.33 | 13.33 | 21.43 | 30.77 | 0 | 0 | 0 |
| | Max | 77.42 | 71.43 | 65.00 | 71.11 | 71.15 | 58.00 | 76.19 | 77.42 |
| | $Q_1$ | 53.85 | 40.83 | 33.10 | 40.00 | 42.11 | 40.53 | 30.00 | 39.34 |
| | Me | 57.52 | 50.00 | 40.28 | 44.09 | 48.48 | 47.81 | 43.59 | 50.00 |
| | $Q_3$ | 68.33 | 58.46 | 46.62 | 51.22 | 54.84 | 50.00 | 48.89 | 52.86 |
| Efficiency Part II | $x$ | 50.10 | 61.72 | 41.38 | 37.36 | 51.00 | 39.10 | 62.04 | 48.43 |
| | N | 10 | 21 | 23 | 21 | 29 | 9 | 8 | 121 |
| | SD | 27.26 | 30.34 | 33.19 | 28.29 | 28.43 | 19.11 | 21.07 | 29.55 |
| | Min | 0.00 | 0.00 | 0.00 | 0.00 | 0.00 | 20.00 | 33.33 | 0.00 |
| | Max | 85.7 | 100.0 | 100.0 | 100.0 | 100.0 | 72.7 | 100.0 | 100.0 |
| | $Q_1$ | 33.33 | 40.00 | 0.00 | 20.00 | 28.57 | 25.00 | 47.32 | 28.57 |
| | Me | 50.00 | 62.5 | 42.86 | 33.33 | 50.00 | 33.33 | 63.33 | 50.00 |
| | $Q_3$ | 80.00 | 83.33 | 75.00 | 50.00 | 75.00 | 44.44 | 70.83 | 72.73 |
| Efficiency Part III | $\bar{x}$ | 55.71 | 44.17 | 48.01 | 26.81 | 49.78 | 52.92 | 65.28 | 46.50 |
| | N | 7 | 16 | 17 | 16 | 26 | 10 | 6 | 98 |
| | SD | 45.41 | 32.03 | 41.88 | 34.85 | 36.31 | 26.36 | 20.99 | 35.92 |
| | Min | 0.00 | 0.00 | 0.00 | 0.00 | 0.00 | 25.00 | 40.00 | 0.00 |
| | Max | 100.0 | 100.0 | 100.0 | 100.0 | 100.0 | 100.0 | 100.0 | 100.0 |
| | $Q_1$ | 0.00 | 16.67 | 0.00 | 0.00 | 25.00 | 33.33 | 50.00 | 0.00 |
| | Me | 50.00 | 50.00 | 50.00 | 8.33 | 50.00 | 50.00 | 63.33 | 50.00 |
| | $Q_3$ | 100.00 | 58.30 | 80.00 | 47.20 | 80.00 | 50.00 | 75.00 | 75.00 |

Comparative analyses found no significant differences in technical–tactical performance (TTP) in relation to the compared weight categories. The exception was the attack efficiency in Part I, where significant variation was noted (Table 7).

**Table 7.** Comparative Summary of Technical–Tactical Indicators for the Three Parts of the Fight with Weight Categories (ANOVA KW Test).

| Indicator | *p* |
|---|---|
| Activity Part I | 0.6860 |
| Activity Part II | 0.1260 |
| Activity Part III | 0.8350 |
| Effectiveness Part I | 0.2181 |
| Effectiveness Part II | 0.3901 |
| Effectiveness Part III | 0.1921 |
| Efficiency Part I | 0.0008 * |
| Efficiency Part II | 0.0643 |
| Efficiency Part III | 0.2320 |
| Effectiveness of Referee Penalties | 0.8401 |

* $p < 0.05$.

It was observed that out of 229 matches, 132 competitors scored a total of 6747 technical points. Statistical analysis showed that competitors most frequently scored points in the first part of the fight, and least frequently in Part III (Table 8). Additionally, the number of points scored in each part differed significantly in statistical terms ($p = 0.0000$).

**Table 8.** Descriptive Statistics of the Total Points Scored in Parts I, II, and III, and the Total Points from All Three Parts.

| Indicator | N | $\bar{x}$ | Me | Min | Max | $Q_1$ | $Q_3$ | SD | V |
|---|---|---|---|---|---|---|---|---|---|
| Total Points Part I | 132 | 23.2 | 20 | 0 | 91 | 10 | 32.5 | 16.7 | 71.8 |
| Total Points Part II | 132 | 5.3 | 4 | 0 | 27 | 2 | 8 | 5.0 | 94.1 |
| Total Points Part III | 132 | 2.3 | 2 | 0 | 20 | 0 | 3.5 | 3.2 | 135.7 |
| Total Points from All Three Parts | 132 | 30.8 | 26.5 | 0 | 107 | 13 | 42 | 21.3 | 69.0 |

In all weight categories, the most points were scored in Part I of the fight (56 kg = 85%, 62 kg = 80%, 69 kg = 77%, 77 kg = 84%, 85 kg = 77%, 94 kg = 78%, +94 kg = 72%). The fewest points were scored in Part III (56 kg = 5%, 62 kg = 7%, 69 kg = 10%, 77 kg = 4%, 85 kg = 9%, 94 kg = 8%, +94 kg = 14%). In all weight categories, the total points scored in the fight ranged from 80% to 51%. The effectiveness of referee penalties for the entire competition was an average level of $1.16 \pm 0.84$ points. The highest values for the effectiveness of referee penalties were noted in the 62 kg category, while the lowest values were recorded in the above 94 kg category (Table 9).

**Table 9.** Effectiveness of Referee Penalties Divided by Weight Categories.

| | | 56 kg | 62 kg | 69 kg | 77 kg | 85 kg | 94 kg | +94 kg | Total |
|---|---|---|---|---|---|---|---|---|---|
| | $\bar{x}$ | 1.19 | 1.36 | 1.09 | 1.05 | 1.11 | 1.32 | 1.00 | 1.16 |
| | N | 10 | 24 | 24 | 22 | 29 | 12 | 11 | 132 |
| | SD | 1.00 | 1.12 | 0.75 | 0.62 | 0.86 | 0.77 | 0.61 | 0.84 |
| Effectiveness of | Min | 0.00 | 0.00 | 0.00 | 0.00 | 0.00 | 0.00 | 0.00 | 0.00 |
| Referee Penalties | Max | 3.50 | 4.00 | 3.33 | 2.00 | 3.00 | 3.00 | 2.00 | 4.00 |
| | $Q_1$ | 0.33 | 0.50 | 0.50 | 0.60 | 0.50 | 0.88 | 0.50 | 0.50 |
| | Me | 1.00 | 1.17 | 1.00 | 1.00 | 1.00 | 1.33 | 1.00 | 1.00 |
| | $Q_3$ | 1.60 | 2.00 | 1.42 | 1.50 | 1.50 | 1.73 | 1.50 | 1.55 |

The total average match time was 256 s and ranged between 55 and 432.5 s. Of this, the effective fight time was 144 s, while the break time amounted to 112 s (Table 10).

**Table 10.** Descriptive Statistics of the Total, Effective, and Break Times for the Entire Group of Competitors.

| Fight Time | N | $\bar{x}$ | Me | Min | Max | $Q_1$ | $Q_3$ | SD | V |
|---|---|---|---|---|---|---|---|---|---|
| Total Time | 132 | 256 | 256 | 55 | 432.5 | 225 | 296 | 61 | 24 |
| Break Time | 132 | 112 | 111 | 5 | 263.5 | 83 | 138 | 43 | 38 |
| Effective Time | 132 | 144 | 146 | 50 | 216 | 129 | 165 | 28 | 19 |

## 4. Discussion

The aim of our study was to characterize Ju-Jitsu sports fighting in the "fighting" formula for all participants in the championship tournament, both globally for the entire competition as well as by individual weight category. In our research, a significant number of Ju-Jitsu bouts ended early due to a Full Ippon (32.31%), although more than half of the tournament confrontations were decided by a technical point advantage (58.52%). Winning before time is the highest form of advantage over an opponent and provides the most satisfaction and glory to the winner. However, when two similarly trained opponents face off and the fight is closely contested, the outcome is often decided by technical points. Based on empirical observation and competitive-coaching experience, one can distinguish fighters who try to impose their style from the first second of the fight, aiming for a quick end, and those who control their opponent for the prescribed duration of the fight, aiming for a points advantage. This is dictated by the individual fighting model and the structural–functional profile of the competitor [24]. This may suggest to coaches a profile for specialized combat

preparation of athletes for subsequent championship-level competitions. In qualitative terms, a warrior should strive to resolve the confrontation early but should also be mentally, volitionally, and endurance-ready to fight the full distance.

Attack activity, according to the original assumptions of combat sports theory, is the sum of executed technical–tactical actions [25]. An example of high activity among fighting competitors is the analysis of kickboxing bouts, which reaches the highest values compared with other indicators [26]. We observed a similar trend in our research, where the attack activity index was highest in the first part of the fight but significantly decreased in the second and third parts. This is a consequence of fighting tactics and the structure of the bout, in which competitors aim to achieve the largest point advantage by dominating the first part of the fight. Striking combat in a standing position is characterized by a high frequency of blows, which is confirmed by numerous analyses of boxing and kickboxing bouts [18,27], the specifics of which correspond to the first part of a Ju-Jitsu bout. A similar trend was observed in terms of attack effectiveness (the ratio of effective techniques to the total number of observed bouts), which was highest in Part I of the duel. This proves that as the activity index for attacks increases, its effectiveness proportionally rises in Part I of the fight. Significant variations in the level of activity and effectiveness in different parts of the fight were shown, further emphasizing the importance of the striking–standing aspect of Ju-Jitsu. The fight begins in this plane and mastering its intricacies is essential for practitioners of this sport to achieve success. In our championship tournament research, the level of fighting in Parts II and III showed significantly lower indicator values compared to strikes. There is a suggestion to enrich training programs focused on increased activity and effectiveness in these parts of the fight. Effective takedowns provide technical points and enable ground fighting, thereby exploiting the possibilities offered by this aspect of combat. Effective Part III techniques have high point values, which may decide the final result [11]. Analysis of weight categories showed that the highest activity in the first part of the fight was in the athletes weighing up to 77 kg, while in the second and third parts of the fight, the highest activity was in the category of athletes weighing up to 85 kg. On the other hand, the lowest activity in the first and second parts was recorded in the heaviest weight category—+94 kg—while in the third part, the lowest activity was recorded in the category of athletes weighing up to 62 kg. Attack effectiveness was highest in the category up to 56 kg in the first part, in the category up to 85 kg in the second part, and in the +94 kg category in the third part. The lowest performance was observed in the +94 kg category in the first part, in the up to 69 kg category in the second part, and in the up to 77 kg category in the third part. Reports from the combat sports community indicate that athletes in heavier-weight divisions throw fewer strikes compared with those in lighter-weight categories; this is associated with structural and functional potential [28]. Our research results partially confirm this, particularly in terms of activity in the first and second parts of the fight and the effectiveness of the first part in the heaviest athletes. Generally, the observed trend of variations in indicators for a given category was not straightforward, meaning it was not directly proportional (higher indicator values vs. higher body mass) or inversely proportional (higher indicator values vs. lower body mass). Qualitatively, the nature of the three-dimensional combat in JU-JITSU (stand-up striking and grappling, as well as ground fighting) and the associated broad concept of specialization in fighting style with the individual preferences of the studied athletes in the sports competition could explain this phenomenon.

The effectiveness index characterizes the quantitative ratio of effective attacks to all executed attacks. Therefore, its high values may coexist with low fighter activity [25]. The average value of attack effectiveness in our study was 47.05% across all parts during the entire championship. Similar trends were observed in each part of the fight. However, when weight categories were considered, the heaviest division, i.e., +94 kg, led globally. In contrast, for Part I, lighter weight classes up to 56 kg and 62 kg showed effectiveness advantages. Lighter fighters are characterized by higher speed, agility, and stamina [29], which may have translated into higher effectiveness in these weight categories. Interestingly,

this index is dominated by the heaviest category, +94 kg, in Parts II and III. Heavier weight classes are characterized by a higher level of muscular strength, which may favor takedowns and ground fighting, thus affecting the effectiveness index [30]. In summary, the comparative analysis of our research revealed no significant differences in technical–tactical performance across the compared weight categories. An exception was found in the effectiveness of the attack in the first part of the fight, where significant variations were observed with respect to weight categories. Our research analysis showed that fighters score the most technical points during the first part of the fight. This is a consequence of the high activity and effectiveness indices observed in this plane of combat. High striking–standing activity in our studies led to higher strike effectiveness and a greater number of technical points. This is associated with the dynamics of the competition, which takes place in a standing position and structurally resembles kickboxing [18]. Individual technical actions or their combinations are characterized by high execution speed, less complex movement structure, and simpler starting positions compared with techniques in Parts II and III of the fight.

An important tactical aspect of the fight is the referee's penalties, given for exceeding the rules, as they provide points to the opponent [5]. Rydzik et al., in their observational studies, identified violations during kickboxing fights, showing that athletes most commonly commit two illegal actions during a match [31]. The effectiveness of referee penalties for the entire competition, based on their research, was at a moderate level of 1.16 points; this varied by weight category. The observed discrepancy may arise from differing knowledge of the rules and the ability to practically apply them under pressure during tournament fights. It could also be due to chance, as Ju-Jitsu fights are characterized by high dynamics.

In their research, the most common penalty was the minor penalty "shido," which affected 89% of athletes, followed by a medium penalty "chui" (39%). Balci and Ceylan proved that the penalty "shido" significantly affects the technique and tactics of athletes, and indirectly affects the outcomes of JUDO competitions [32]. Molina et al. demonstrated that receiving a "shido" is directly associated with the outcome of the JUDO competition, increasing the likelihood of loss [33].

In top-level competitions, only 2% of athletes were disqualified (Hansoku-Make), indicating a good understanding of the regulations and the ability to adhere to them in order to avoid the harshest penalty, which results in immediate defeat [5].

Regarding violations, the majority of athletes (83%) were found to be lacking in Part II of the fight. This could be justified by the structure of the match and the tactical actions of the athletes, who avoid fighting in Part II due to its effectiveness in terms of point scoring or a lack of competence in this area [34]. Sequentially, based on the level of participation, athletes committed violations such as uncontrolled strikes, lack of fighting in Part I, strikes after a clinch, lack of fighting in Part III, and holding the head. It is recommended that comprehensive training practice content that shapes and improves knowledge, views, and practical functioning be introduced into the sphere of regulation compliance and tournament atmosphere. This type of action requires the organization of refereed, simulated sparring during training sessions. During the referee's ruling on effective techniques (scored) and regulatory violations, it is recommended that the sparring be paused for a detailed analysis and assessment of the executed action by all trainees.

To become accustomed to the tournament atmosphere, which is different from the training environment, it is recommended that a designated fighting area, a referee, and an audience made up of all trainees and possibly visiting guests be organized during training.

Previous studies showed that periods of intense effort and breaks were observed in Ju-Jitsu sports fights. The total fight time ranged from 55 to 432.5 s, with an average time of 256 s. This result indicates a large discrepancy in the total fight time. Various factors influenced this observation. The shortening of the struggle time was due to fights ending before time due to achieving three Ippons in different parts of the bout, or due to injury or disqualification. The extended time was due to breaks caused by bodily injuries,

regulatory requirements, or tactical procedures (e.g., adjusting the athlete's attire). On the other hand, the effective fighting time was much shorter, averaging 144 s. Effective fighting is a reflection of the athlete's offensive capability, which may influence the outcome of the bout [35].

An innovative value of the conducted analysis is the introduction of an assessment based on technical–tactical indicators, in individual parts of the fight, which provides the possibility of a comprehensive and precise evaluation of the athletes' actions in hand-to-hand combat during sports competitions. This type of action constitutes a credible identification of the baseline situation and provides a foundation for potential targeted training intervention.

*Limitations of the Study*

Our research had certain limitations. Specifically, we only examined elite-level tournament athletes. In future comparative studies, it is recommended that the sample be expanded to include athletes of national championship or local tournament status for more comprehensive comparisons. Another limitation was the lack of consent to expand the scope of research during the competitions, which precluded the assessment of adaptive effort parameters and the psychological potential of the athletes. To capture a multi-dimensional clinical context, it is advisable to broaden the diagnostic approach to include measurements such as blood lactate concentration, stress levels, motivation, aggression, and mental attitude, and to explore their associations with technical–tactical preparation.

Furthermore, for a thorough understanding of the multi-dimensional clinical context, future research should aim to identify model characteristics of medalists at this level of competition. Additionally, it should be possible to differentiate and analyze, in detail, the sub-skills of technical abilities in different parts of the fight in future studies.

## 5. Conclusions

Analysis and evaluation of results from the conducted observations allowed for the following conclusions:

1. It was shown that over half of the sports fights (58.52%) ended in a win through technical point advantage within the full timeframe of the bout, whereas 32.31% of fights ended with a Full Ippon victory.
2. The highest level of activity and attack effectiveness was observed in Part I of the fight, whereas the lowest was observed in Part III; furthermore, significant variations were shown across different parts of the fight. Middleweight fighters displayed the highest activity in the first part of the fight, while heavyweight categories dominated in the remaining parts of the bout.
3. The level of attack effectiveness throughout the competition showed similar values in each part of the fight, except for weight categories in Part I, where the highest values were recorded in the light categories. The values for the medium and heavy categories were similar.
4. Athletes in the heavier weight category displayed higher levels of effectiveness and attack efficiency in Parts II and III, whereas the opposite trend was observed for the lightest weight categories in Part I.
5. The highest level of technical points scored was recorded in Part I of the fight, and significant variations were again noted across the parts.
6. Athletes were most frequently penalized with a minor penalty called shido; only 2% received a disqualification (hansoku make), with the most frequent violation being a lack of fighting in Part II.
7. The total fight time was 256 s, of which 144 s were effective fighting and 112 s were breaks.

*Practical Application*

The study shows that effective JU-JITSU competition requires comprehensive motor, technical–tactical, and mental preparation. Athletes should focus on striking skills to gain technical points and on mastering throws and takedowns for potential high-point gains in later parts of the fights. Simulated sparring is advised to get accustomed to tournament conditions. Strength-speed and endurance-strength exercises, such as interval and circuit training, are recommended for conditioning. The use of Technical–Tactical Indicators (TTI) can optimize training by allowing diagnosis and intervention in various aspects of fighting, benefiting not just JU-JITSU but also other combat sports.

**Author Contributions:** Conceptualization, T.A., Ł.R., A.K. and Z.M.; methodology, T.A., Ł.R. and A.K.; software, T.A., Ł.R., A.K. and Z.M.; validation, T.A., Ł.R. and A.K.; formal analysis, T.A., Ł.R. and A.K.; investigation, T.A., Ł.R. and A.K.; resources T.A., Ł.R., A.K. and D.A.; data curation, T.A., Ł.R., A.K. and D.A.; writing—original draft preparation, T.A., Ł.R., A.K. and W.W.; writing—review and editing, T.A., Ł.R., A.K., W.J.C. and W.W.; visualization, T.A., Ł.R. and A.K.; supervision, T.A., Ł.R., A.K. and W.W.; project administration, T.A., Ł.R., A.K. and W.J.C.; funding acquisition, T.A., Ł.R. and A.K. All authors have read and agreed to the published version of the manuscript.

**Funding:** This research received no external funding.

**Institutional Review Board Statement:** The study was conducted in accordance with the Declaration of Helsinki and approved by the Ethics Committee of the University of Rzeszów (protocol code 8 December 2021).

**Informed Consent Statement:** Informed consent was obtained from all subjects involved in the study.

**Data Availability Statement:** The data presented in this study are available upon request from the corresponding author.

**Conflicts of Interest:** The authors declare no conflict of interest.

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
