# Peer review of "Analysis of Combat in Sport JU-JITSU during the World Championships in Fighting Formula"

_applsci, doi:10.3390/app132011417_

Round 1
Reviewer 1 Report
The aim of the study was a general and detailed analysis of sport fighting in Ju Jitsu during a high level competition. The study provided a comprehensive analysis of multiple fights, showing the different strategies adopted by the competitors. I only have some minor issues that I would like the authors to address.
Line 49: Authors should refrain from using the term "phase" as competitors are allowed to switch their fighting method at any moment in time. It is not like during a sprint event where we know that during the first 0-40m is acceleration phase, 40-60m is max speed phase etc. For ju jitsu there isn't a fixed time point.
Line 69: "plane" should it be "plan"?
Line 104: Please define "PPT".
Line 164-228: How did the authors came up with these formula? Is there any reference?
Author Response
Dear Reviewer,
Thank you very much for your time and valuable comments, which all have been considered and incorporated. The detailed list of responses is given below. We hope that the modifications and explanation will be acceptable for you.
Yours sincerely,
Rydzik, corresponding author
The aim of the study was a general and detailed analysis of sport fighting in Ju Jitsu during a high level competition. The study provided a comprehensive analysis of multiple fights, showing the different strategies adopted by the competitors. I only have some minor issues that I would like the authors to address.
Line 49: Authors should refrain from using the term "phase" as competitors are allowed to switch their fighting method at any moment in time. It is not like during a sprint event where we know that during the first 0-40m is acceleration phase, 40-60m is max speed phase etc. For ju jitsu there isn't a fixed time point.
A: Thank you for your good attention, we have changed the spelling to Part, in line with international regulations fighting (JJIF). https://jjif.sport/
Line 69: "plane" should it be "plan"?
A: This has been correct
Line 104: Please define "PPT".
A: This has been correct
Line 164-228: How did the authors came up with these formula? Is there any reference?
A: Added references under indicators [19,25,26]
Reviewer 2 Report
Many thanks to the authors for the manuscript.
However, I would like to draw attention to the observed shortcomings and share my comments.
Conclusions (rows 31-33) presented in the work abstract are not related to the content of the work, they must be corrected.
The authors pay little attention to the differentiation of the studied data according to the weight categories of the subjects. That is a critical research shortcoming. Making generalizations without considering the fighters' weight classes is wrong in my opinion. Please refer to the authors of this manuscript: Santos, M. A. F. D., Soto, D. A. S., de Brito, M. A., Brito, C. J., Aedo-Muñoz, E., Slimani, M., ... & Miarka, B. (2023). Effects of weight divisions in time-motion of female high-level Brazilian Jiu-jitsu combat behaviors. Frontiers in Psychology, 14, 477.
I think it is necessary to differentiate the insights made taking into account whether the athletes belong to the light, medium and heavyweight categories.
I missed the limitation and Strengths sections of the study. I think they are necessary.
After the corrections have been made, I suggest that the manuscript be accepted for publication.
Sincerely.
Author Response
Dear Reviewer,
Thank you very much for your time and valuable comments, which all have been considered and incorporated. The detailed list of responses is given below. We hope that the modifications and explanation will be acceptable for you.
Yours sincerely,
Rydzik, corresponding author
Conclusions (rows 31-33) presented in the work abstract are not related to the content of the work, they must be corrected.
A: This has been corrected
The authors pay little attention to the differentiation of the studied data according to the weight categories of the subjects. That is a critical research shortcoming. Making generalizations without considering the fighters' weight classes is wrong in my opinion. Please refer to the authors of this manuscript: Santos, M. A. F. D., Soto, D. A. S., de Brito, M. A., Brito, C. J., Aedo-Muñoz, E., Slimani, M., ... & Miarka, B. (2023). Effects of weight divisions in time-motion of female high-level Brazilian Jiu-jitsu combat behaviors. Frontiers in Psychology, 14, 477.
A: We find it difficult to agree as most of the results are given by weight category see table 1, table 4, 5, 6 , 9. The temporal structure, on the other hand, see Table 10 , refers to the entire course of the tournament. Consequently, it is difficult to make a weight division. In addition, there are 7 weight categories in juitsu , and they are not referred to as light , medium and heavy. The article you refer to is about movement behaviour in a fight and we refer the indicators for this type of behaviour in our article to the weight categories. However, in line with your comment, we add references of % values in each weight category for points obtained in a fight.
I think it is necessary to differentiate the insights made taking into account whether the athletes belong to the light, medium and heavyweight categories.
A: This has been corrected
I missed the limitation and Strengths sections of the study. I think they are necessary.
After the corrections have been made, I suggest that the manuscript be accepted for publication.
A: Thank you
Round 2
Reviewer 2 Report
I thank the authors for their efforts.
Although the authors present information differentiated by weight categories in the tables, the purpose of the study, data analysis, discussion, and conclusions provide summarized data and conclusions.
In my opinion, this goes against the logic of martial arts.
Also, the authors do not present a research limitation. Submitting the limitation is necessary in cases like the one presented in the study. See: Ross PT, Bibler Zaidi NL. Limited by our limitations. Perspect Med Educ. 2019 Aug;8(4):261-264. doi: 10.1007/s40037-019-00530-x. PMID: 31347033; PMCID: PMC6684501.
Sincerely.
Author Response
Dear Reviewer,
Thank you very much for your time and valuable comments, which all have been considered and incorporated. The detailed list of responses is given below. We hope that the modifications and explanation will be acceptable for you.
Yours sincerely,
Rydzik, corresponding author
I thank the authors for their efforts.
Although the authors present information differentiated by weight categories in the tables, the purpose of the study, data analysis, discussion, and conclusions provide summarized data and conclusions.
In my opinion, this goes against the logic of martial arts.
A: Added information on weight categories in the executive abstract, discussion, conclusions and objective
Also, the authors do not present a research limitation. Submitting the limitation is necessary in cases like the one presented in the study. See: Ross PT, Bibler Zaidi NL. Limited by our limitations. Perspect Med Educ. 2019 Aug;8(4):261-264. doi: 10.1007/s40037-019-00530-x. PMID: 31347033; PMCID: PMC6684501.
A: Added limitations to the study